# Intermedin 1-53 Ameliorates Atrial Fibrosis and Reduces Inducibility of Atrial Fibrillation via TGF-β1/pSmad3 and Nox4 Pathway in a Rat Model of Heart Failure

**DOI:** 10.3390/jcm12041537

**Published:** 2023-02-15

**Authors:** Shenzhou Ma, Feng Yan, Yinglong Hou

**Affiliations:** 1Shandong Provincial Qianfoshan Hospital, Cheeloo College of Medicine, Shandong University, Jinan 250100, China; 2Cardiology Departments, The First Affiliated Hospital of Shandong First Medical University, Jinan 250014, China; 3Department of Emergency Medicine, Qilu Hospital, Cheeloo College of Medicine, Shandong University, Jinan 250100, China

**Keywords:** atrial fibrosis, atrial fibrillation, upstream therapy, intermedin 1-53, heart failure

## Abstract

Objective: New drugs to block the occurrence of atrial fibrillation (AF) based on atrial structural remodeling (ASR) are urgently needed. The purpose of this study was to study the role of intermedin 1-53 (IMD1-53) in ASR and AF formation in rats after myocardial infarction (MI). Material and methods: Heart failure was induced by MI in rats. Fourteen days after MI surgery, rats with heart failure were randomized into control (untreated MI group, n = 10) and IMD-treated (n = 10) groups. The MI group and sham group received saline injections. The rats in the IMD group received IMD1-53, 10 nmol/kg/day intraperitoneally for 4 weeks. The AF inducibility and atrial effective refractory period (AERP) were assessed with an electrophysiology test. Additionally, the left-atrial diameter was determined, and heart function and hemodynamic tests were performed. We detected the area changes of myocardial fibrosis in the left atrium using Masson staining. To detect the protein expression and mRNA expression of transforming growth factor-β1 (TGF-β1), α-SMA, collagen Ⅰ, collagen III, and NADPH oxidase (Nox4) in the myocardial fibroblasts and left atrium, we used the Western blot method and real-time quantitative polymerase chain reaction (PCR) assays. Results: Compared with the MI group, IMD1-53 treatment decreased the left-atrial diameter and improved cardiac function, while it also improved the left-ventricle end-diastolic pressure (LVEDP). IMD1-53 treatment attenuated AERP prolongation and reduced atrial fibrillation inducibility in the IMD group. In vivo, IMD1-53 reduced the left-atrial fibrosis content in the heart after MI surgery and inhibited the mRNA and protein expression of collagen type Ⅰ and III. IMD1-53 also inhibited the expression of TGF-β1, α-SMA, and Nox4 both in mRNA and protein. In vivo, we found that IMD1-53 inhibited the phosphorylation of Smad3. In vitro, we found that the downregulated expression of Nox4 was partly dependent on the TGF-β1/ALK5 pathway. Conclusions: IMD1-53 decreased the duration and inducibility of AF and atrial fibrosis in the rats after MI operation. The possible mechanisms are related to the inhibition of TGF-β1/Smad3-related fibrosis and TGF-β1/Nox4 activity. Therefore, IMD1-53 may be a promising upstream treatment drug to prevent AF.

## 1. Introduction

Atrial fibrillation (AF) is a supraventricular tachyarrhythmia with uncoordinated atrial activation and, consequently, ineffective atrial contraction. The characteristics on an electrocardiogram (ECG) include (1) irregular R–R intervals (when atrioventricular (AV) conduction is present), (2) the absence of distinct repeating P waves, and (3) irregular atrial activity [1]. AF is an irregular and often rapid heart rate. As a clinical sustained cardiac arrhythmia and a common complication of heart failure (HF) [2], AF causes enormous economical and clinical burdens [3,4]. Currently, traditional antiarrhythmic drugs show two main limitations [5]: (1) safety concerns about “on-target” and “off-target” adverse effects, and (2) a limited curative effect due to the biological nature of ion channel remodeling. Therefore, in the early stage of atrial remodeling, upstream drugs targeting potential arrhythmogenic substrate should theoretically improve this situation.

Atrial structural remodeling (ASR) is the arrhythmogenic substrate in the onset, lasting, and recurrence of AF [6]. As a sign of atrial remodeling, atrial fibrosis is the morphological basis of atrial fibrillation. Transforming growth factor β1 (TGF-β1) and Nox4 have a vital function in orchestrating the process of fibrogenesis [7,8], which could accumulate collagen deposition, mediate the excessive proliferation of atrial fibroblasts, and aggravate ASR, eventually leading to AF formation. Thus, inhibiting TGF-β1 and Nox4 is likely to inhibit atrial fibrosis and suppress the occurrence and maintenance of AF.

Intermedin (IMD), also known as adrenomedullin 2 (ADM2), was identified by Roh [9] and Kobayashi [10] at almost the same time. The prepropeptide of 148 amino acids is encoded by the human IMD gene, including an N-terminus signal peptide for secretion. IMD1-53 is produced by the precursor IMD through proteolysis. As a vital bioactive peptide, IMD1-53 is widely expressed in the whole body and is highly expressed in the heart and vasculature [11]. It maintains vascular homeostasis and has powerful cardiovascular protective functions, such as cardiac protection, the regulation of blood pressure, the acceleration of angiogenesis, and anti-apoptosis. In addition, IMD1-53 has an anti-fibrosis effect on lung [12] and renal fibrosis [13], and other studies have demonstrated that IMD1-53 can ameliorate fibrosis of the left ventricle (LV) in animal models of hypertension [14], MI [15], and LV hypertrophy [16]. IMD1-53 is an effective anti-fibrosis hormone that inhibits cardiac fibrosis formation after MI by downregulating the expression of TGF-β and the phosphorylation of smad3 [17]. In another study, intermedin alleviated unilateral ureteral obstruction (UUO)-induced renal fibrosis by the inhibition of Nox4 [18]. IMD1-53 is highly expressed in the atrium of the MI-induced heart failure rat model [19]. However, we know little about the effect of IMD1-53 on atrial fibrosis and AF formation in the process of HF after the MI operation. The hypothesis is that IMD1-53 ameliorates atrial fibrosis and reduces the inducibility of AF through inhibiting TGF-β/psmad3 and Nox4. The purpose of this study was to study the effects of IMD1-53 on atrial fibrosis and AF formation in rats with HF after MI.

## 2. Materials and Methods

### 2.1. Peptide Synthesis and Reagents

Human IMD (IMD1-53) with the sequence His–Ser–Gly–Pro–Arg–Arg–Thr–Gln–Ala–Gln–Leu–Leu–Arg–Val–Gly–Cys–Val–Leu–Gly–Thr–Cys–Gln–Val–Gln–Asn–Leu–Ser–His–Arg–Leu–Trp–Gln–Leu–Met–Gly–Pro–Ala–Gly–Arg–Gln–Asp–Ser–Ala–Pro–Val–Asp–Pro–Ser–Ser–Pro–His–Ser–Tyr–NH_2_ with an intramolecular disulfide bond between Cys16 and Cys21 was synthesized by ShineGene Bio-Technologies (Shanghai, China) [16].

TGF-β1 (PeproTech, London, UK), primary antibodies against GAPDH (Beyotime, Beijing, China), TGF-β1 (Proteintech, Wuhan, China), α-SMA (Proteintech, Wuhan, China), Collagen Ⅰ (Abcam, Cambridge, UK), Collagen III (Abcam, Cambridge, UK), Nox4 (Sigma, Saint Louis, MO, USA), and Smad3/pSmad3 (Sigma, Saint Louis, MO, USA), as well as HRP-conjugated secondary antibody (Beyotime, Beijing, China), SB431542 (Darmstadt, Germany), and sodium pentobarbital (Sinopharm Chemical Reagent Co., Ltd., Shanghai, China) were used. 

### 2.2. Establishment of Animal Models

All animals received humane care, and the research procedures were carried out in accordance with the approved protocols and guidelines formulated by the Animal Care Committee of Shandong Qianfoshan Hospital (S337). MI surgery was conducted after a 7 day adaptation period.

Male Wistar rats (Vitalriver Company, Beijing, China) weighing 250–300 g (6–8 weeks old) were housed under a 12:12 h light/dark cycle at 21 °C, 50% humidity. The rats had free access to tap water and rat chow. All rats were anesthetized with 3% sodium pentobarbital (Sinopharm Chemical Reagent Co., Ltd.) at 3 mg/100 g of body weight intraperitoneally, and then intubated and ventilated by air through a ventilator (Harvard, Rodent Ventilator, model 683, with a frequency of 60 ventilations per minute and a volume of 1.5 mL/100 g). Before disinfection with 2% *w*/*v* chlorhexidine gluconate in 70% *v*/*v* isopropyl alcohol, the chests of the rats were shaved. After left thoracotomy and pericardiotomy, 6.0 polypropylene suture was used to ligate the left anterior descending artery at approximately 2 to 3 mm from its starting point. Ten rats underwent a sham operation after pericardiotomy, but the coronary artery was not ligated. Echocardiography was performed after 2 weeks of the MI surgery. According to A. Martin Gerdes’s experience and literature reports [20], to develop the MI heart failure rat model, a large myocardial infarction area is needed to increase AF inducibility. Therefore, only rats with large infarction areas (>40% of LV circumference in short-axis view [21]) were selected for further experiments in this study. The MI rats were randomized into two groups (using a table of random digits): an untreated MI group (MI; n = 10) and an IMD1-53-treated group (IMD; n = 10) that received IMD1-53 at 10 nmol/kg/day intraperitoneally [19] for 4 weeks. The sham group only received a saline injection.

### 2.3. Echocardiography and Hemodynamic Measurement

Four weeks after the intraperitoneal administration of IMD1-53, all rats were lightly anesthetized with 1% sodium pentobarbital at 3 mg/100 mg (Sinopharm Chemical Reagent Co., Ltd.), and transthoracic echocardiography was performed with a 30 MHz high-frequency transducer (Vevo2100; VisualSonics, Inc., Toronto, ON, Canada). To calculate the left-ventricular ejection fraction (LVEF) and left-ventricular fractional shortening (LVFS), the left-ventricular diameter at the end of diastole and end of systole was measured using M-mode tracing [21,22]. After the echocardiographic study, the hemodynamic parameters were measured using the BL-420E biological system (Chengdu Tai-Meng Science and Technology Co., Ltd., Chengdu, Sichuan, China) [20]. Briefly, a catheter was inserted into the left ventricle through the right common carotid artery to record the left-ventricular end-diastolic pressure (LVEDP) [23].

### 2.4. Electrophysiology Study and Atrial Fibrillation Inducibility Test

A 1.6F octopolar Millar electrophysiology catheter (EPR-802; Millar Instruments, Houston, Texas) was used to assess the cardiac electrophysiology. Briefly, the eight-pole catheter was inserted into the right atrium, and the atrial electrogram was recorded through the right jugular vein. The PowerLab data acquisition systems (JinJiang Electronic Science and Technology, Sichuan, China) were used to record right-atrial electrocardiograms and standard surface electrocardiographic lead II from three pairs of electrodes. The atrial electrograms from the proximal, middle, and distal pairs were recorded to determine the atrial capturing and atrial fibrillation pattern. Standard S1S2-programmed pacing protocols and regular pacing were used to determine the atrial effective refractory period (AERP). The longest coupling interval without capturing the atria’s electrical activity was defined as AERP. The right atria were paced at a cycle length of 120 ms (three thresholds) [20]. To induce AF, burst pacing at 50 Hz was used. For each rat, the average duration of arrhythmia based on five such tests was calculated. Then, the duration of the spontaneous AF burst pacing was recorded.

As mentioned earlier, AF is defined as irregular rapid atrial activation with different electrographic patterns lasting for 0.5 s [24]. The atrial rates in rats with atrial fibrillation were 1500 beats/min. Some rats with induced atrial arrhythmias had regular atrial electrographic patterns, and the rates (ranging from 900 to 1300 beats/min) were lower than those in AF. These arrhythmias are similar to patients with atrial tachycardia or atrial fibrillation. We combined those arrhythmias and typical atrial fibrillation as a single entity of atrial tachyarrhythmias.

### 2.5. Preparation of Atrial Samples

After conducting the electrophysiology study and atrial fibrillation inducibility test, all rats were euthanized with 3% pentobarbital sodium (3 mg/100 mg; Sinopharm Chemical Reagent Co., Ltd.), and their hearts were removed. The left atria were removed and cut into two pieces. One piece was frozen in liquid nitrogen and maintained at −80 °C for both mRNA and protein analysis. The other piece was paraffin-embedded for Masson’s trichrome staining.

### 2.6. Culture and Identification of Cardiac Fibroblasts

The hearts of 1–3-day-old Wistar rats were removed, and their membrane envelopes were cut. To obtain the atrium, the apex of the heart was removed. The atria were cut into 0.5–1.0 mm^3^ fragments and digested with 0.1% collagenase Ⅱ (Gibco, Waltham, MA, USA). The pieces of atria were cultured in an incubator at 5% CO_2_ and 37 °C for 24 h; then, through differential adhesion, the cardiac fibroblasts were obtained [25]. The second to fourth generations of cardiac fibroblasts were selected for the following experiments. The fibroblast medium consisted of Dulbecco’s modified Eagle medium (DMEM, Gibco, USA), 1% PS (Gibco, USA), and 10% fetal bovine serum (FBS, Gibco, USA). The atrial fibroblast fibrosis model was induced by TGF-β1 (PeproTech, London, UK) for 24 h. The cells were randomly divided into one of the following six groups: control (untreated cells), TGF-β1 (cells were stimulated with TGF-β1 at 10 ng/mL), TGF-β1 + IMD1-53 (cells were cultured with IMD1-53 at 1 × 10^−7^ mmol/L and stimulated with TGF-β1), SB431542 (SB431542 was added to the medium for 1 h at 2 μΜ), TGF-β1 + SB431542, and TGF-β1 + SB431542 + IMD1-53.

### 2.7. RNA Isolation and Amplification

TRIzol reagent (Invitrogen, Carlsbad, CA, USA) was used to extract the total RNA from the left-atrial tissue according to the manufacturer’s instructions. The RNA was reverse-transcribed into cDNA using oligo-dT15 primers and M-MLV reverse transcriptase (TIANGEN, Beijing, China). Quantitative real-time polymerase chain reaction (RT-qPCR) [26] was used to measure the levels of desmoplastic factor expression of TGF-β1, Nox4, α-SMA, collagen Ⅰ, and collagen III by using an ABI ViiA 7 Real-Time PCR system (Applied Biosystems, Foster City, CA, USA). PCR was performed using the following primers: glyceraldehyde 3-phosphate dehydrogenase (GAPDH) (sense): 5′–GAAGGTGGTGAAGCAGGCATCC–3′ and (antisense): 5′–GGCACTGTTGAAGTCGCAGGAG–3′, TGF-β1 (sense): 5′–CCAGATCCTGTCCAAACTAAGG–3′ and (antisense): 5′–CTCTTTAGCATAGTAGTCCGCT–3′, α-SMA(sense): 5′–GACGCTGAAGTATCCGATAGAA–3′ and (antisense): 5′–AATACCAGTACGTCCAGAG–3′, collagen I (sense): 5′–TGAACGTGGTGTACAAGGTC–3′ and (antisense): 5′–CCATCTTTACCAGGAGAACCAT–3′, collagen III (sense): 5′–GAAAGAATGGGGAGACTGGAC–3′ and (antisense): 5′–TACCAGGTATGCCTTGTAATCC–3′, Nox4 (sense): 5′–TTCTGGACCTTTGTGCCTATAC–3′ and (antisense): 5′–CCATGACATCTGAGGGATGATT–3′. All primers were designed according to their published sequences using the GenScript online primer design software and synthesized by Sigma-Genosys. Each gene was analyzed in triplicate. The relative quantification of expression was performed using the 2^−ΔΔCT^ method [27] with GAPDH for normalization.

### 2.8. Immunoblot

RIPA lysis buffer (Beyotime, Beijing, China) was used to extract the total protein from frozen atrial tissues or atrial fibroblasts. The BCA Protein Assay Kit (Beyotime, Beijing, China) was used to quantify the protein concentration. Then, 10% sodium dodecyl sulfate polyacrylamide gel electrophoresis (SDS-PAGE) was used to separate the protein samples (50 μg), and the protein samples were subsequently electro-transferred to the PVDF membranes [28]. The PVDF membranes with the protein samples were incubated overnight at 4 °C with diluted primary antibodies against GAPDH (1:2000, Beyotime, Beijing, China), TGF-β1 (1:3000, Proteintech, Wuhan, China), α-SMA (1:800, Proteintech, Wuhan, China), Collagen Ⅰ (1:1000, Abcam, Cambridge, UK), collagen III (1:5000, Abcam, Cambridge, UK), Nox4 (1:1000, Sigma, Saint Louis, MO, USA), or Smad3/pSmad3 (1:1000, Sigma, Saint Louis, USA), followed by incubation with an HRP-conjugated secondary antibody (1:10,000, Beyotime, Beijing, China). ImageJ 1.46r software (NIH Image) was used to quantitate the antibody-positive proteins on the PVDF membranes by measuring the band intensity (Areax OD) and normalizing to GAPDH intensity. By standardizing the data to the control values, the results are expressed as folded changes.

### 2.9. Masson Staining of Heart Tissue

The left-atrial samples were fixed for 24 h with 4% paraformaldehyde. Masson’s trichrome staining was performed after standard paraffin embedding. The Sens Viewer software (Olympus Imaging Corp., Tokyo, Japan) was used to evaluate the left-atrial fibrosis area (%) in each group by calculating the total left-atrial fibrosis.

### 2.10. Data Analysis and Statistics

Data were expressed as the means ± SD, except the atrial fibrillation duration, which was expressed as the median and IQR (interquartile range). Fisher’s exact test was used to compare atrial fibrillation inducibility between the MI group and the IMD1-53 group. Comparisons among groups were performed with ANOVA followed by Bonferroni’s post-test, and the least significant difference method was used for post hoc multiple comparisons. All statistical analyses used SPSS 17.0 software (IBM Corp., Armonk, NY, USA). A two-tailed *p*-value < 0.05 was considered statistically significant.

## 3. Results

### 3.1. Effects of IMD1-53 Treatment on Body Weight

During the 4 weeks treatment period, no mortality was recorded. The body weight was comparable among the groups; there was no significant difference across the three groups (Table 1).

### 3.2. Effects of IMD1-53 on LV Hemodynamics

Compared with the sham rats, the LVEDP in the MI group was increased (Table 1), whereas IMD1-53 treatment decreased the LVEDP compared to the MI group (*p* < 0.01), indicating an improved diastolic LV function after IMD1-53 treatment.

### 3.3. IMD1-53 Improved Cardiac Function and Reduced Left-Atrial Diameter after MI

Figure 1A shows the representative M-mode echocardiograms of each group. The LVEF and LVFS in the MI group were significantly lower than those in the sham group, while they were higher in the IMD1-53 group than in the MI group (*p* < 0.05; Figure 1C,D). LAD, an indicator of left-atrial remodeling, was higher in the MI group compared with the sham group (*p* < 0.01; Figure 1B). IMD1-53 treatment decreased LAD (*p* < 0.01; Figure 1B). These results showed that IMD1-53 could inhibit the cardiac dysfunction caused by MI and reduce atrial dilation.

### 3.4. Effects of IMD1-53 on Cardiac Electrophysiology and Atrial Fibrillation Inducibility

The heart rate of the MI rats was significantly higher than that of the sham rats (*p* < 0.05, Table 1), and it was inhibited by IMD treatment. In addition, the AERP was significantly shorter in the MI group compared with the sham rats (*p* < 0.01) and significantly longer compared with the MI group (*p* < 0.01, Figure 2B). Figure 2A shows an example of the original electrocardiographic of one rat with atrial fibrillation induced by S1S2 stimulation. In total, 6/10 rats in the MI group induced atrial tachyarrhythmia (atrial fibrillation in three rats, atrial tachycardia/fibrillation in two rats, and arrhythmia in one rat). In contrast, 3/10 rats in the IMD group induced atrial fibrillation (*p* < 0.05), while the sham operated rats did not induce any arrhythmias (Figure 2C). The induced atrial fibrillation duration was significantly shorter in the IMD group than in the MI group (Figure 2D).

### 3.5. IMD1-53 Inhibited Left-Atrial Fibrosis

Masson’s trichrome-stained heart sections confirmed that MI led to increased fibrosis in the left atrium. The area of left-atrial fibrosis increased significantly in the MI group compared to the sham group, and IMD1-53 treatment decreased the areas of fibrosis (*p* <0.01; Figure 3). Type Ⅰ and III collagen mRNA expression in the left-atrial tissue increased in the MI group. This increase was significantly inhibited after IMD1-53 treatment (*p* < 0.05, *p* < 0.01, Figure 4A,B). In addition, the protein levels of type Ⅰ and III collagen in the IMD1-53 group were significantly lower than those in the MI group (*p* < 0.05, *p* < 0.01, Figure 4C–E), which indicated that IMD1-53 inhibited left-atrial fibrosis.

### 3.6. IMD1-53 Inhibited Differentiation of Cardiac Fibroblasts to Myofibroblasts

α-SMA is a marker that fibroblasts differentiate into myofibroblasts. Therefore, we tested the mRNA and protein expression of α-SMA to assess the role of IMD1-53 in the differentiation of fibroblasts to myofibroblasts. In the MI group, both the mRNA and the protein expression of α-SMA was higher than that in the sham group. However, α-SMA expression was significantly lower in the IMD group compared to the MI group (*p* < 0.05; Figure 5B,F). These results indicated that IMD1-53 inhibited atrial fibroblast differentiation.

### 3.7. IMD1-53 Inhibited Expression of TGF-β1, NOX4, and Smad3 Phosphorylation in the Left Atrium

The TGF-β1/Smad3 signaling pathway is essential for the progression of fibrosis. In the present study, we explored whether IMD1-53 could inhibit fibrosis through the TGF-β1 signaling pathway. RT-qPCR and Western blot analysis suggested that the expression of TGF-β1 was significantly higher in the MI group than in the sham group and lower in the IMD group (*p* < 0.05; Figure 5A,E). Western blot analysis indicated that Smad3 phosphorylation was significantly higher in the MI group than in the sham group, and Smad3 phosphorylation was inhibited by IMD1-53 (*p* < 0.05; Figure 5H,I). In addition, the mRNA and protein expression of Nox4 was significantly increased in the MI group, and both mRNA and protein were markedly inhibited by IMD1-53 (*p* < 0.05; Figure 5C,G).

### 3.8. IMD1-53 Inhibits Nox4 Upregulation via TGF-β1/ALK5-Dependent Pathway in Atrial Fibroblast

It was aimed to determine whether the inhibition role of IMD1-53 on Nox4 expression is mediated via the TGF-β1/ALK5-dependent pathway; the atrial fibroblasts treated with TGF-β1 showed a significant increase in Nox4 expression to control cells (Figure 6). Consistent with the results in vivo, IMD1-53 significantly reduced TGF-β1-stimulated Nox4 upregulation. To determine whether the ALK5 receptor was involved in the IMD1-53-inhibited Nox4 expression, the atrial fibroblast was pretreated with the ALK inhibitor SB431542 for 1 h. The treatment with SB431542 at 2 μΜ abolished the Nox4 upregulation induced by TGF-β1. On this basis, IMD1-53 could further reduce the protein expression of Nox4 compared to the TGF-β1 + IMD cell group. This result demonstrates that IMD1-53 partially inhibits Nox4 upregulation via the TGF-β1/ALK5 pathway.

## 4. Discussion

We investigated the effect of IMD1-53 on MI heart failure-induced atrial fibrosis and atrial fibrillation inducibility. The main results of this study are as follows: (1) IMD1-53 treatment increased AERP, reduced atrial fibrillation inducibility, and shortened the atrial fibrillation duration in rats with MI; (2) IMD1-53 treatment reduced LAD and ameliorated cardiac function; (3) IMD1-53 treatment inhibited left-atrial fibrosis and collagen production; (4) IMD1-53 impacted TGF-β1/Smad3, and the TGF-β1/ALK5/Nox4 pathway inhibited atrial fibrosis.

The remodeling of atrial structure, electrophysiology, and intracellular Ca^2+^ handling after heart failure increases susceptibility to atrial fibrillation in patients and animal models [29]. Heart failure induced by MI is an essential cause of atrial fibrillation; thus, we chose the MI disease model [20]. Atrial fibrillation requires a substrate to initiate. Atrial fibrosis is a crucial component of the arrhythmogenic substrate to increase electrical remodeling and promote atrial fibrillation in the MI models. The remodeling processes in the MI-induced HF model increased LV systolic dysfunction, which led to atrial wall pressure overload, promoting the formation of arrhythmogenic substrate. Subsequently, the MI-induced HF rats exhibited an increased inducibility to AF. On the basis of the above standpoints, anti-fibrosis is considered a primary or secondary measure for AF prevention in AF upstream therapy [29,30].

Atrial fibrosis is a hallmark of atrial structural remodeling, with a complex multifactorial process, including differentiation of fibroblasts tagged by α-SMA and irregular deposition of ECM proteins, such as collagen types I and III [31]. Fibroblast differentiation leads to an arrhythmogenic atrial substrate because of an unbalance in the synthesis and degradation of ECM proteins [32]. This study showed that the collagen fiber volume was significantly increased by Masson’s staining, and both mRNA and the protein levels of α-SMA and collagen type Ⅰ and III increased in the MI rats. However, IMD1-53 treatment decreased the collagen fiber volume and expression of collagen type Ⅰ and III as well as α-SMA. Our findings showed that IMD1-53 can inhibit atrial fibrosis and limit the excessive proliferation of myofibroblasts in some way.

Compared to the sham group, the MI heart failure rats developed significant left-atrial diameter enlargement, fibrosis, fibroblast differentiation, and increased AF vulnerability. Other researchers also found similar results [21]. However, the IMD1-53 treatment significantly reduced atrial fibrosis, which was associated with reduced vulnerability to atrial fibrillation. These results suggest that IMD1-53 may be a promising bioactive substance and can be used as an upstream treatment for AF in heart failure after the MI process. Several plausible mechanisms may be the basis for IMD1-53 to block the development of arrhythmogenic substrate for AF.

Previous studies have found that TGF-β1 overexpression in animal models [33,34,35] and AF patients [29,30] will lead to increased cardiac fibrosis, cardiac displacement, cardiac dysfunction, and fibrosis-related arrhythmia. Suppressing TGF-β1 can effectively inhibit atrial fibrosis and reduce the susceptibility to AF [33,36,37,38]. The TGF-β1 pathways include typical Smad-dependent and non-canonical Smad-independent pathways, which promote cardiac fibroblast differentiation into myofibroblasts and the synthesis of collagen fibers by cardiac fibroblasts [39]. In the canonical Smad-dependent pathway, TGF-β binds to ALK5 and type II TGF-β receptor (Tβ RⅡ), forming the Smad 2/3/4 complex, which then leads to Smad protein-mediated signal transduction [40,41]. Therefore, the TGF β1/Smad3 pathway may be a promising target for anti-fibrosis. Our study showed that IMD1-53 downregulated TGF β1/Smad3 phosphorylation after MI surgery.

Increasing evidence shows that the NADPH oxidase-dependent redox signaling involved in the profibrotic responses is mediated by TGF-β [42]. There are five subtypes of Nox catalytic subunits, namely, Nox1, Nox2, Nox3, Nox4, and Nox5 [43]. Current evidence suggests that the Nox4 type, NADPH oxidase, may have an essential role in mediating TGF-β-induced profibrotic responses [41,44,45,46].Jarman et al. showed that compound 88 (an oral Nox4 inhibitor) can reduce bleomycin-induced pulmonary fibrosis in rats and TGF-β-induced procollagen and α-SMA expression in human lung fibroblasts [47]. This study suggests that the inhibition of the Nox4 function may be a novel treatment in anti-fibrosis treatment. In our study, we proved that IMD1-53 inhibited the mRNA and protein expression of Nox4 after MI operation. In vitro, TGF-β1 stimulated Nox4 upregulation, and IMD1-53 inhibited the protein expression of Nox4.

ALK5 is one of the most important TGF-β receptor types. SB-431542 is a potent and specific inhibitor of ALK5 [48]. We confirmed that the expression of TGF-β1 and collagen I and III was significantly inhibited by IMD1-53, thus demonstrating the ability of IMD1-53 to inhibit atrial fibrosis. We also found that, after inhibiting ALK5 by SB431542 in the atrial fibroblast, IMD1-53 could further lower the protein expression of Nox4 compared to the group inhibited by SB431542 only. A possible mechanism is that IMD1-53 inhibited Nox4 not only via TGF-β1/ALK5, but also through another signal pathway.

Part of the benefit of suppressing AF by IMD1-53 may be attributable to the improvement of heart function. Clinical and animal data all indicate that MI-induced heart failure has LA hemodynamic overload and atrial remodeling, which can increase the risk of AF [33]. Accordingly, the treatment of improved heart function itself may reduce the inducibility to AF. Consequently, the reduced AF vulnerability by IMD1-53 may benefit from improved heart function. In this study, we delivered IMD1-53 4 weeks post MI, which would, in part, provide a cardioprotective effect and contribute to reduce atrial remodeling in the long term. Previous research has also noted the cardioprotective effect of IMD1-53 [10]. Consequently, further investigation into the effect of different IMD1-53 treatment timings on AF is needed.

## 5. Conclusions

IMD1-53 reduced the inducibility and duration of AF and atrial fibrosis in the rat after MI operation. The plausible mechanisms are associated with the suppression of TGF-β1/Smad3-related fibrosis and TGF-β1/Nox4 activity. Therefore, IMD1-53 may be a promising active agent as upstream therapy for the prevention of AF after MI.

## Figures and Tables

**Figure 1 jcm-12-01537-f001:**
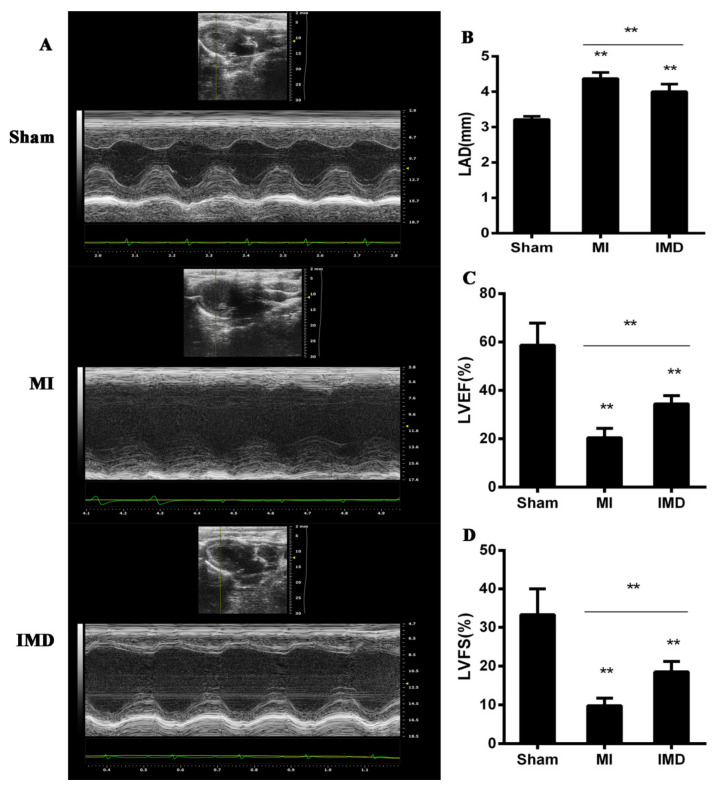
Effects of IMD1-53 treatment on cardiac contractile function and LAD after MI surgery. (**A**) Representative examples of M-mode echocardiogram of three groups. (**B**) Bar graphs indicate the LAD. IMD1-53 decreased LAD after MI. (**C**) Bar graphs indicate the LVEF. IMD1-53 improved LVEF after MI. (**D**) Bar graphs indicate the LVFS. IMD1-53 improved LVFE after MI. Data in bar graphs are means ± SD, n = 10. LVEF, left-ventricular ejection fraction; LVFS, left-ventricular fraction shortening; LAD, left-atrial diameter; MI, myocardial infarction; IMD1-53, intermedin 1-53. ** *p* < 0.01.

**Figure 2 jcm-12-01537-f002:**
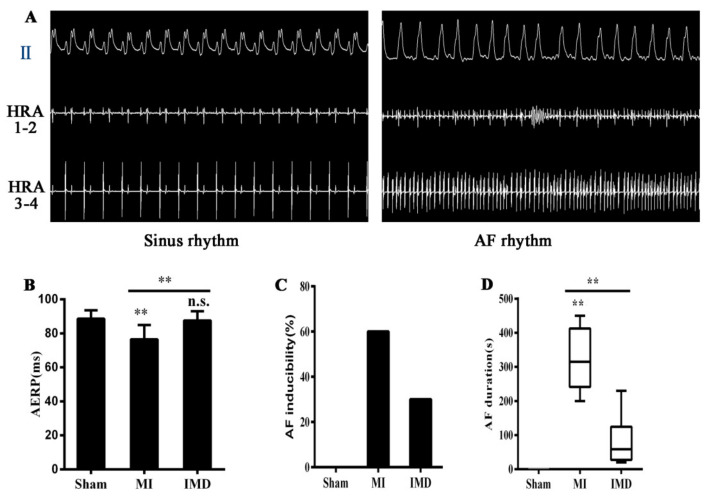
Characteristics of cardiac electrophysiology in rats. (**A**) Electrocardiogram of SR and AF. Right-atrial electrograms (HRA1-2 and HRA3-4) demonstrate rapid irregular atrial activations with varying electrogram morphology. Note that different atrial electrical activation patterns were recorded from the high right atrium (HRA1-2) and lower right atrium (RA3-4) during sinus rhythm and atrial fibrillation in rats. (**B**) Changes of AERP in rats. IMD1-53 improved AERP after the MI operation. (**C**) Bar graphs indicate the AF inducibility. IMD1-53 inhibited AF inducibility compared to the MI group. (**D**) Bar graphs indicate the AF duration. IMD1-53 decreased AF episode duration. Data in bar graphs are means ± SD, except the bar graph of figure (**D**), which presents the median and IQR (25–75%), n = 10. AF, atrial fibrillation. HRA, high right atrium; RA, right atrium. ** *p* < 0.01.

**Figure 3 jcm-12-01537-f003:**
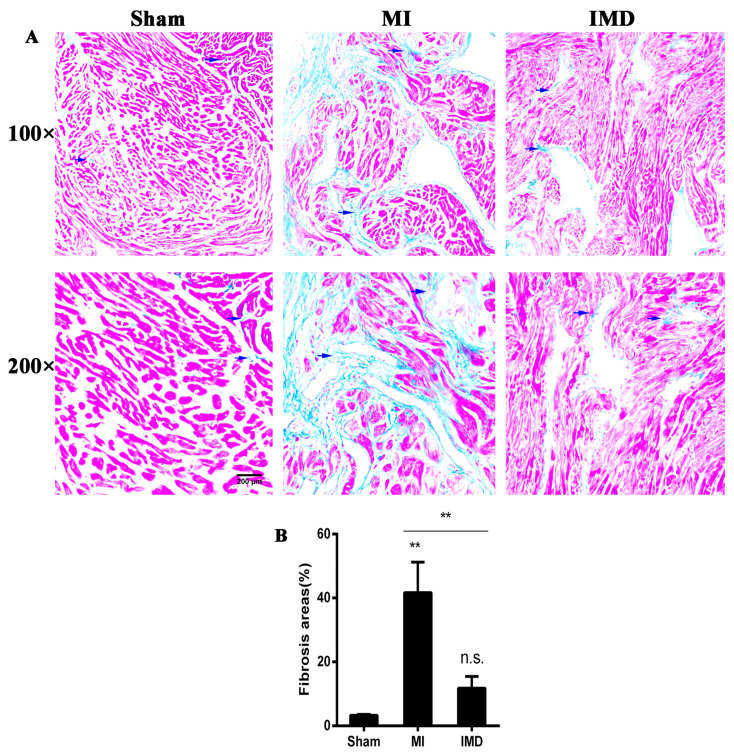
Effects of IMD-1-53 treatment on atrial fibrosis after MI surgery. (**A**) Representative photomicrographs of left-atrial histologic slides (Masson trichrome stain) from one rat in each group are shown at the top. The blue arrow indicates fibrosis. Scale bar = 200 μm. (**B**) Values are presented as means ± SD. IMD1-53 reduced fibrosis area. n = 10. MI, myocardial infarction. ** *p* < 0.01, n.s., no significance.

**Figure 4 jcm-12-01537-f004:**
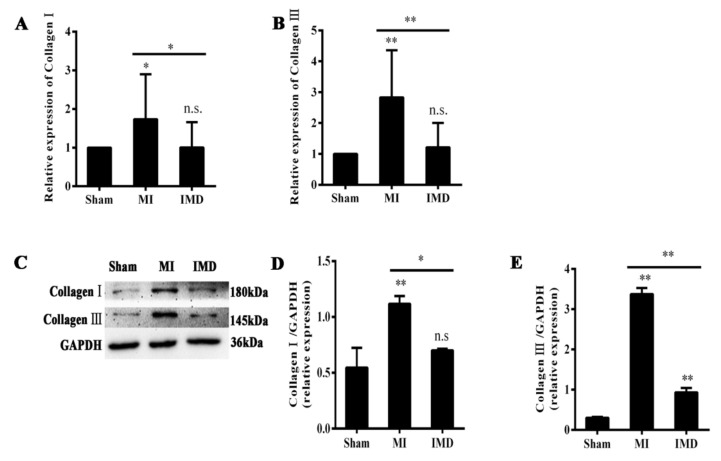
mRNA and protein expression of collagen I and collagen III in the left atria of rats. (**A**) mRNA expression of collagen I. IMD1-53 decreased collagen I after MI. (**B**) mRNA expression of collagen III. IMD1-53 decreased collagen III after MI. (**C**) Western blot of collagen I and collagen III. (**D**) Bar graphs indicate Western blot analysis for the protein expression of collagen I. IMD1-53 decreased the protein level of collagen I in the left atrium. (**E**) Bar graphs indicate Western blot analysis for the protein expression of collagen III. IMD1-53 decreased the protein level of collagen III in the left atrium. Data in bar graphs are means ± SD, n = 10. * *p* < 0.05, ** *p* < 0.01. n.s., no significance.

**Figure 5 jcm-12-01537-f005:**
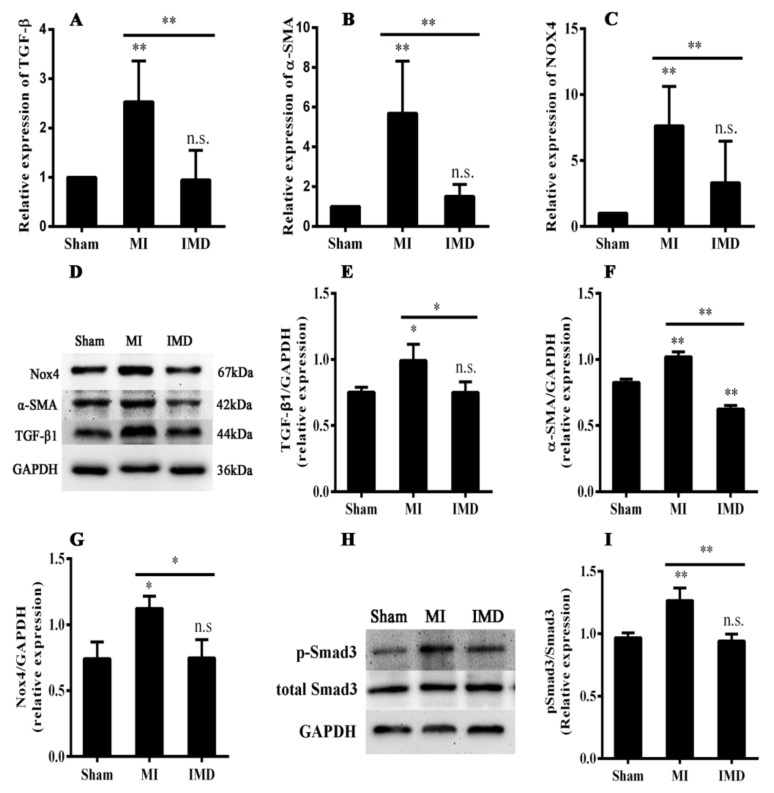
mRNA and protein expression of TGF-β1, α-SMA, Nox4, and Smad3 in the left atria of rats. (**A**) mRNA expression of TGF-β1 in rat’s left atrium. IMD1-53 reduced TGF-β1 after MI. (**B**) mRNA expression of α-SMA in rat’s left atrium. IMD1-53 reduced α-SMA after MI. (**C**) mRNA expression of Nox4 in rat’s left atrium. IMD1-53 reduced Nox4 after MI. (**D**–**G**) Western blot analysis for the protein expression of TGF-β1, α-SMA, and Nox4. IMD1-53 decreased each of these indices. (**H**,**I**) Representative Smad3 protein expression and phosphorylation level of Smad3 measured by Western blot in left atrium. IMD1-53 reduced the phosphorylation of Smad3 after MI. Data in the bar graphs are means ± SD, n = 10. * *p* < 0.05, ** *p* < 0.01. n.s., no significance.

**Figure 6 jcm-12-01537-f006:**
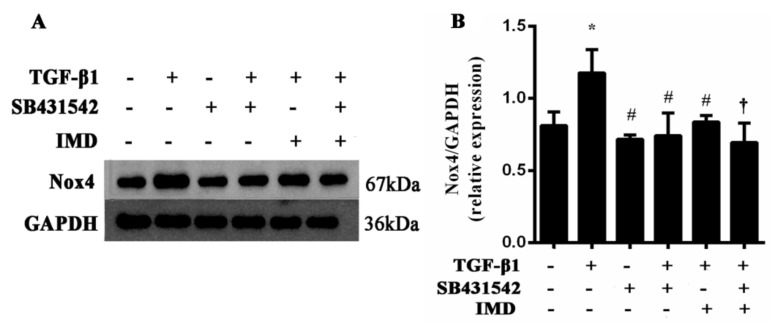
TGF-β1-induced Nox4 expression is inhibited by IMD via TGF-β1/ALK5-dependent pathway. (**A**) Representative Nox4 protein expression measured by Western blot in atrial fibroblast. (**B**) Densitometric quantifications of band intensities from Western blot for Nox4/GAPDH in atrial fibroblast. Data in bar graphs are means ± SD, n = 10. * *p* < 0.05 versus the control group, ^#^
*p* < 0.05 versus the TGF-β1 group, ^†^
*p* < 0.05 versus the TGF-β1 + IMD group.

**Table 1 jcm-12-01537-t001:** Details for body weight (BW), heart rate (HR), and LVEDP.

	Sham	MI	IMD
Body weight (g)	331 ± 37.3	351 ± 30.6 ^n.s.^	345 ± 30.5 ^n.s.^
Heart rate (beats/min)	361 ± 48.0	419 ± 15.8 *	386 ± 28.2 ^n.s.^
LVEDP (mmHg)	14 ± 3.7	30 ± 6.7 **	19 ± 8.3 ^#^

Values are presented as the mean ± SD. * *p* < 0.05 vs. the sham rats, ** *p* < 0.01 vs. the sham rats, ^#^
*p* < 0.05 vs. the MI rats; n.s., no significance. LVEDP, left-ventricular end-diastolic pressure; MI, myocardial infarction; IMD, intermedin 1-53.

## Data Availability

Data are available on request from the corresponding author.

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
