# Peer review of "Intermedin 1-53 Ameliorates Atrial Fibrosis and Reduces Inducibility of Atrial Fibrillation via TGF-β1/pSmad3 and Nox4 Pathway in a Rat Model of Heart Failure"

_jcm, 2023, doi:10.3390/jcm12041537_

Round 1
Reviewer 1 Report
The paper titled “Intermedin1-53 ameliorates atrial fibrosis and reduces inducibility of atrial fibrillation via TGF-β1/pSmad3 and Nox4 path- 2 way in a rat model of heart failure”. The study aimed to investigate the effects of IMD1-53 on atrial 72 fibrosis and AF formation in rats with HF after MI. the manuscript was written in a good manner and the methods used are well described . but need some improvement:
- Introduction : Smad and ALK pathway should be mentioned in details to illustrates the mechanism of Intermedin1-53
Material and methods:
- chemicals used should be mentioned
- pre-treated with the ALK inhibitor SB431542 not mentioned
-Echocardiography and hemodynamic measurement: need reference of the method
PCR : - a reference or the methods used for primers sequence should be mentioned.
- reference for “2-ΔΔCT method is needed
-. Culture of cardiac fibroblasts: a reference should be added
-line 157: :” IMD” should be corrected to “IMD1-53”
- “IMD1-53”; the abbreviation should be revised in the whole manuscript to be the same.
Immunoblotting : a Reference should be added
- Line 224 & line 335:” IMD” should be corrected to “IMD1-53”
- Figure 4; needs to be improved with good resolution, the quality of western blot photo needs to be improved.
- In Figure 2 (D) ; Authors should add the type of statistical test used for atrial fibrillation duration analysis in the figure ligand.
- Discussion: add paragraph about the Intermedin1-53 action, its receptors and their effect on Smad/ALK signaling
Author Response
- Introduction : Smad and ALK pathway should be mentioned in details to illustrates the mechanism of Intermedin1-53
Smad and ALK pathway now are mentioned in “Introduction”.
Material and methods:
- chemicals used should be mentioned
The chemicals are mentioned in the manuscript are as follows:
TGF-β1 (PeproTech, London, UK) ,Primary antibodies against GAPDH (Beyotime, Beijing, China), TGF-β1(Proteintech, Wuhan, China), α-SMA(Proteintech, Wuhan, China), Collagen Ⅰ (Abcam, Cambridge, UK), Collagen Ⅲ (Abcam, Cambridge, UK) or, Nox4(Sigma, Saint Louis, USA), Smad3/pSmad3(Sigma, Saint Louis, USA) and HRP-conjugated secondary antibody (Beyotime, Beijing, China), SB431542 (Darmstadt, Germany), sodium pentobarbital (Sinopharm Chemical Reagent Co., Ltd.).
- pre-treated with the ALK inhibitor SB431542 not mentioned
ALK inhibitor SB431542 was added to the medium for 1h at 2 μΜ.
-Echocardiography and hemodynamic measurement: need reference of the method
The reference has been mentioned in manuscript. (Ma J, Yin C, Ma S, Qiu H, Zheng C, Chen Q, Ding C, Lv W.Shensong Yangxin capsule reduces atrial fibrillation susceptibility by inhibiting atrial fibrosis in rats with post-myocardial infarction heart failure. Drug Des Devel Ther. 2018; 12:3407-18.)
PCR : - a reference or the methods used for primers sequence should be mentioned.
- reference for “2-ΔΔCT method is needed
-. Culture of cardiac fibroblasts: a reference should be added
- Immunoblotting : a Reference should be added
All those four references have been mentioned in the article.
-line 157: :” IMD” should be corrected to “IMD1-53”
- “IMD1-53”; the abbreviation should be revised in the whole manuscript to be the same.
- Line 224 & line 335:” IMD” should be corrected to “IMD1-53”
All “IMD” has been corrected to “IMD1-53”
- Figure 4; needs to be improved with good resolution, the quality of western blot photo needs to be improved.
The resolution of Figure 4 have been improved.
- In Figure 2 (D); Authors should add the type of statistical test used for atrial fibrillation duration analysis in the figure ligand.
Data in bar graphs of “Figure 2 (D)” is the median and IQR (25%–75%). Comparisons among groups were performed with ANOVA followed by Bonferroni’s post-test.
- Discussion: add paragraph about the Intermedin1-53 action, its receptors and their effect on Smad/ALK signaling on Smad/ALK signaling
ALK5 is one of the most important TGF-β receptor types. SB-431542 is a potent and specific inhibitor of ALK5. We confrmed that the expression of TGF-β1, collagen Ⅰ and Ⅲ were signifcantly inhibited by IMD1-53, thus demonstrating the ability of IMD1-53 to inhibit atrial fibrosis. We also found that after inhibiting the ALK5 by SB431542 in atrial fibroblast, IMD1-53 can further lower the protein expression of Nox4 compared to the only SB431542 inhibited group. The possible mechanism is that IMD1-53 inhibited Nox4 not only via TGF-β1/ALK5, but also through other signal pathway.
Reviewer 2 Report
The authors provide an interesting experiment using Intermedin1-53 as a regulator of atrial fibrosis and atrial fibrillation induction, maybe giving the rational base for human trials in a critical topic (cardiac remodeling and arrhythmia generation). The work is also well-referenced.
However, I've got some concerns about the quality of the presentation of both methods and results.
First of all, the overall text is often hard to read. I suggest English revision.
In the introduction, definitions of human heart failure and atrial fibrillation should be revised, they appeared too superficial. Furthermore, JCM Is a clinical journal and most readers are clinicians that are not so familiar with arrhythmias in animal models. you reported a graphical representation in the results, which I think is more opportune in this introduction (even a theoretical scheme, you can maintain your registration in the results). This is referred also for section 2.4
Section 2.2 should be rewrite. I think the subgroup division should be reported before the description of the intervention. How did you establish the number of needed rats in every group? It is totally arbitral, or there was similar studies with similar numbers (add eventually references).
There are some normal ranges for echocardiographic (and not also echocardiographic) measures? You substantially consider normal the ranges of the shame intervention group, but if there are some universally accepted ranges are useful in the correct interpretation of the overall results.
Why did you use mean and SD for all the variables except for AF duration? As a very small sample, the median and IQR appeared more correct for all the continuous variables. Even a better description of categorical variables and why you choose fisher exact test. Kruskal-wallis test appeared more appropriate than standard anova in non-parametrical/small groups statistics.
Table 1 in particular but also multiple descriptions of figures are unclear, in particular, the description of p values is very dispersive. Try to rewrite table 1 and insert in the figures or in the text better the p values
You also used expressions for p values as < 0,01, < 0,05. In particular, the latter is not correct. You should choose a number of significant figures (4 or 5) and adopt for all the p values, writing the less approximative value possible, in particular for those near 0,05.
Phrases as "echocardiography...intraperitoneally" and "masson...deposition" are more methods than results. Revise both sections to improve homogeneity and reduce repetitions.
Frequently, in the results, I didn't find analysis (or didn't find very clear and complete of p values) between treated rats and shame rats. Please add differences, it's interesting to understand if the effect of the drug reconducts to normality, or some degree of atrial fibrosis and arrhythmogenic risk persists.
Even how did you represented the variables in graphs is more a method than a result. Write in the statistical analysis even the graphical interpretation, so you don't have to rewrite it for every single figure.
Did The study have limitations? Did you perform some bias assessment?
Best regards
Author Response
Table 1 in particular but also multiple descriptions of figures are unclear, in particular, the description of p values is very dispersive. Try to rewrite table 1 and insert in the figures or in the text better the p values
This has been rewrite.
You also used expressions for p values as < 0,01, < 0,05. In particular, the latter is not correct. You should choose a number of significant figures (4 or 5) and adopt for all the p values, writing the less approximative value possible, in particular for those near 0,05.
This has been revised.
Phrases as "echocardiography...intraperitoneally" and "masson...deposition" are more methods than results. Revise both sections to improve homogeneity and reduce repetitions.
This has been revised.
Frequently, in the results, I didn't find analysis (or didn't find very clear and complete of p values) between treated rats and shame rats. Please add differences, it's interesting to understand if the effect of the drug reconducts to normality, or some degree of atrial fibrosis and arrhythmogenic risk persists.
These have been revised.
Even how did you represented the variables in graphs is more a method than a result. Write in the statistical analysis even the graphical interpretation, so you don't have to rewrite it for every single figure.
This has been revised.
Did The study have limitations? Did you perform some bias assessment?
The limitations have been mentioned in the article.
(1).Part of the benefits of suppressing AF of IMD1-53 may be attributable to the improvement of heart function. Clinical and animal data all indicate that MI-induced heart failure has LA hemodynamic overload and atrial remodeling which can increase the risk of AF [26]. Accordingly, treatment of improved heart function itself may reduce the inducibility to AF. Consequently, the reduced AF vulnerability by IMD1-53 may be benefited from an improved heart function. In this study, we delivered IMD1-53 four weeks post-MI, which would in part provide a cardioprotective effect and contribute to reduced atrial remodeling in the long term. Not to mention IMD1-53 still have a cardioprotective effect according to a previous study [9]. Consequently, a further investigation of the effect of different treatment timing of IMD1-53 on AF is needed.
(2) We also found that after inhibiting the ALK5 by SB431542 in atrial fibroblast, IMD1-53 can further lower the protein expression of Nox4 compared to the only SB431542 inhibited group. The possible mechanism is that IMD1-53 inhibited Nox4 not only via TGF-β1/ALK5, but also through other signal pathway.
We used the tool of Systematic Review Center for Laboratory Animal Experimentation (SYRCLE’s tool) to evaluate the risk of bias in animal experimentations. The results of item 1,2,4,5,6,7,8 ,10 are “YES”. Item 3 is “NO”. Item 9 is “not clear”.
Thank you for your review.
Round 2
Reviewer 2 Report
I'm afraid that all my concerns aren't resolved.
My first comment was about the definitions of atrial fibrillation in humans and rats, as JCM is a clinical journal. Human afib needs to be defined according to guidelines. More details in differentiation (better a figure) between rats' rhythms are also needed.
You didn't provide any information on the presumptive number of rats for the experiment, and also the number of treated rats (even those that didn't result in a large infarct) is needed.
There are some normal ranges for echocardiographic (and not also echocardiographic) measures? If You substantially consider normal the ranges of the shame intervention group, you must specify it. But if there are some universally accepted ranges, I think they are useful in the correct interpretation of the overall results.
Why did you use mean and SD for all the variables except for AF duration? As a very small sample, the median and IQR appeared more correct for all the continuous variables. Even a better description of categorical variables and why you choose fisher exact test. Kruskal-wallis test appeared more appropriate than standard anova in non-parametrical/small groups statistics. I need an explanation.
This phrase: "Western blot analysis and RT-qPCR were used to evaluate the role of IMD1-53 in the differentiation 335 of fibroblasts into myofibroblasts. α- SMA is a marker that fibroblasts differentiate into 336 myofibroblasts. Myocardial infarction group α- The mRNA and protein expression of SMA was 337 higher than that of sham operation group" is more a method rather than a result.
I appreciate the efforts provided in improving the test and the changes provided according to the other comments, but I need answers, even if no changes will be made.
Author Response
- My first comment was about the definitions of atrial fibrillation in humans and rats, as JCM is a clinical journal. Human afib needs to be defined according to guidelines. More details in differentiation (better a figure) between rats' rhythms are also needed.
For human: Atrial fibrillation (AF) is a supraventricular tachyarrhythmia with uncoordinated atrial activation and consequently ineffective atrial contraction. Characteristics on an electrocardiogram (ECG) include 1) irregular R-R intervals (when atrioventricular [AV] conduction is present), 2) absence of distinct repeating P waves, and 3) irregular atrial activity.
For rats: AF is defined as irregular rapid atrial activation with different electrographic patterns lasting for 0.5 seconds. The atrial rates in rats with atrial fibrillation were1,500 beats /min. Some rats induced atrial arrhythmias had regular atrial electrographic patterns, and the rates (ranging from 900 to 1,300 beats /min) were lower than those in AF. These arrhythmias are similar to patients with atrial tachycardia or atrial fibrillationThese arrhythmias were similar to patients with atrial tachycardia or atrial flutter. We combined those arrhythmias and typical atrial fibrillation as a single entity of atrial tachyarrhythmias.
An example of an induced AF episode. Before the burst, the rat was in SR. After termination of the burst (b), the rat displayed
an irregular atrial rhythm with an irregular ventricular response. After a few seconds (c), the AF episode terminated spontaneously and SR resumed. (d) An example of anoninduced AF episode. After termination of the burst, the rat also displayed SR. (e) Esophageal ECG which recorded clear, high, and bidirectional atrial wave that was used to help to determine the position of the electrode and distinguish SR or AF. (f) A pause representing the termination and cardioversion of AF recorded by esophageal ECG (Ma J, Yin C, Ma S, et al. Shensong Yangxin capsule reduces atrial fibrillation susceptibility by inhibiting atrial fibrosis in rats with post-myocardial infarction heart failure. Drug Des Devel Ther. 2018. 12: 3407-3418.).
2.You didn't provide any information on the presumptive number of rats for the experiment, and also the number of treated rats (even those that didn't result in a large infarct) is needed.
At the beginning of the experiment, the presumptive number of rats was 80. For the reason was that the mortality of the MI-rats was 50% in the past study, but in my experiment the mortality of MI-HF rats was 60-70%. Two weeks after MI operation, a doctor specializing in echocardiography underwent echocardiogram for the experiment rats, the rats didn’t result in a large infarct had been euthanized with 3% pentobarbital sodium (the rats was 8).
There are some normal ranges for echocardiographic (and not also echocardiographic) measures? If You substantially consider normal the ranges of the shame intervention group, you must specify it. But if there are some universally accepted ranges, I think they are useful in the correct interpretation of the overall results.
No,there are not some normal ranges for echocardiographic measures in rats. But we considered the LVEF>50% was the normal range of the sham group. Refer to the other papers,the LVEF of sham group is >50%, and the MI-HF group is <50%( Figure is as follow. Martinez PF, Okoshi K, Zornoff LA, et al. Echocardiographic detection of congestive heart failure in postinfarction rats. J Appl Physiol (1985). 2011. 111(2): 543-51.). The results are the same as ours.
Why did you use mean and SD for all the variables except for AF duration? As a very small sample, the median and IQR appeared more correct for all the continuous variables. Even a better description of categorical variables and why you choose fisher exact test. Kruskal-wallis test appeared more appropriate than standard anova in non-parametrical/small groups statistics. I need an explanation.
For AF duration test, after burst pacing, the rats with most severe heart failure can induce lasting atrial fibrillation. This is an extreme variable value. So I used median and IQR for AF duration. Other variables were used mean and SD.
The number of the experiment rats <40, so I chose the Fisher’s exact instand of Chi-Square Tests. The possibility of atrial fibrillation inducibility of the sham group is close to zero. So I only compare the atrial fibrillation inducibility between the MI group and the IMD1-53 group.
I agree that Kruskal-wallis test appeared more appropriate than standard anova in non-parametrical/small group’s statistics. The experimental data were homogeneity of variance test and normality test. So I chose standard anova.
This phrase: "Western blot analysis and RT-qPCR were used to evaluate the role of IMD1-53 in the differentiation 335 of fibroblasts into myofibroblasts. α- SMA is a marker that fibroblasts differentiate into 336 myofibroblasts. Myocardial infarction group α- The mRNA and protein expression of SMA was 337 higher than that of sham operation group" is more a method rather than a result.
This paragraph has been revised.
I appreciate the efforts provided in improving the test and the changes provided according to the other comments, but I need answers, even if no changes will be made.
Thank you for your advices.
